# The Spatio-Temporal Evolution of Food Production and Self-Sufficiency in China from 1978 to 2020: From the Perspective of Calories

**DOI:** 10.3390/foods12050956

**Published:** 2023-02-23

**Authors:** Jing Zhang, Yu Fang, Hua Zheng, Shenggen Fan, Taisheng Du

**Affiliations:** 1Center for Agricultural Water Research in China, China Agricultural University, Beijing 100083, China; 2State Key Laboratory of Urban and Regional Ecology, Research Center for Eco-Environmental Sciences, Chinese Academy of Sciences, Beijing 100085, China; 3College of Economics and Management, China Agricultural University, Beijing 100083, China

**Keywords:** food calories, supply–demand equilibrium, spatio-temporal evolution, pattern, China

## Abstract

Ensuring national food security is an eternal topic. We unified six categories of food with calorie content including grain, oil, sugar, fruits and vegetables, animal husbandry, and aquatic products on the basis of provincial-level data, and we dynamically evaluated caloric production capacity and the supply–demand equilibrium under the increase in feed-grain consumption as well as the food losses and waste in China from 1978 to 2020 at four different levels. The results show that: (1) From the perspective of food production, the total national calorie production showed a linear growth trend, with a growth rate of 31.7 × 10^12^ kcal/year, of which the proportion of grain crops has always exceeded 60%. Most provinces showed significant increasing trends in food calorific production, except for Beijing, Shanghai, and Zhejiang, which showed slightly decreasing trends. The distribution pattern of food calories and their growth rate were high in the east and low in the west. (2) From the perspective of the food supply–demand equilibrium, the national food calorie supply has been in surplus since 1992, but significant spatial heterogeneity is detected, with the Main Marketing Region changing from a tight balance to a short surplus, North China always remaining in calorie shortage, and 15 provinces still presenting supply and demand gaps up to 2020, necessitating the establishment of a more efficient and faster flow and trade system. (3) The national food caloric center has shifted 204.67 km to the northeast, and the population center has shifted to the southwest. The reverse migration of the centers of food supply and demand will further aggravate the pressure on water and soil resources and cause higher requirements for ensuring the circulation and trade system of food supply. The results are of great significance for the timely adjustment of agricultural development policies, making rational use of natural advantages and ensuring China’s food security and sustainable agricultural development.

## 1. Introduction

Since the reform and opening up, China has made remarkable achievements in eradicating hunger through the improvement of agricultural productivity, with the incidence of food deficiency having fallen to below 2.5% in 2019 [1]. However, internal factors such as rapid population growth, water and soil resource constraints [2,3], and environmental carrying capacity constraints [4] coupled with external risks such as China–US trade frictions [5] and the COVID-19 pandemic [6] have continuously influenced the agricultural system and will place further pressure on future food security in China. Within the context of the current carbon-neutral development goals and the post-epidemic era, improving food productivity and ensuring sufficient food supply can facilitate the efficient use of natural resource endowments, rationalize the layout of food production, and reduce unnecessary emissions and resource waste. Studying the spatial patterns of food production and the supply–demand equilibrium over a long time period is of great practical significance for a comprehensive understanding of China’s food security.

The existing literature on food security has mostly focused on grain crops in a narrow sense and analyzed the dynamics of grain production patterns and the evolution of food supply–demand at different scales such as provinces [7], geographical divisions [8], agricultural zones, and north–south divisions [9]. These studies concluded that the center of gravity of grain in China has shifted northwards, and grain production has changed from a “south–north” to a “north–south” pattern. However, the increase in the economic level of a country or region will inevitably lead to a diversified diet structure [10]. From a broad perspective of food security, only by considering various types of food such as cereals, oils, meat, eggs, and dairy products can a more comprehensive picture of food production capacity be obtained. In order to unify different foods, scholars have converted various types of food into raw grains using grain conversion rates [11] and grain equivalence coefficients [12] or into calories and protein from a nutritional perspective. The steady increase in the consumption of animal foods in the diet structure [13] and the correspondingly increasing demand for feed grains have led to an increase in the per capita grain requirement, yet the per capita calorie intake is mostly considered as steady [14]. Therefore, a calorie-based approach allows for a more comprehensive unification of different foods and provides a more scientific foundation to further analyze and compare food production capacity across space and time. The analysis of production and of supply and demand in terms of calories has become a widely used method. Beltran et al. [14] analyzed global food supply and demand and projected future scenarios through the calorie conversion of major grain crops. Tilman [15] calculated the net global demand for crops in terms of calories and protein. However, these large-scale studies generally deal only with major grain crops; this is inadequate, since the diet of the population has become more abundant.

The term “food supply and demand” describes a dynamic relationship between food production and consumption on a certain spatial and temporal scale. A timely and accurate grasp of the dynamic characteristics of food supply and demand is an essential reference for adjusting the structure of food production and formulating relevant agricultural policies. The indicators commonly used to analyze the balance between food supply and demand include the self-sufficiency rate [15,16], per capita grain possession, the food security index, and the feedable population [7,17,18]. Among these, the feedable population is often used to analyze the production capacity of food calories in a region and characterize the surplus and the deficit in calorie supply and demand. Yet, studies have mostly considered either dynamic trends [19] or spatial distribution characteristics [8,18], and those exploring both spatial and temporal patterns in food calorie supply and demand are limited to grains [14] or other crops.

Despite the significance of trade for food supply, the localization of supply chains has been largely emphasized owing to the obstruction and even collapse of global and regional trade amid the COVID-19 pandemic. Considering the carbon-neutrality goal, the localization of supply can also have a positive effect on reducing greenhouse gas emissions during transport and storage [20]. Analyzing local food supply and demand and demonstrating the food self-sufficiency of the region can thus contribute to a better understanding of the current status of, and trends in, agricultural production and regional food supply. In addition, the loss and waste of feed grain and food are important components of the food supply chain deserving attention. On the one hand, the changing dietary structure of the population has led to a continuous increase in feed grain use and a growing conflict between humans and animals for food [21,22], indicating that feed grain is becoming an important post-harvest flow of food [23]. Huang et al. [22] evaluated China’s feed grain demand and concluded that the feed grain deficit will further increase. On the other hand, about one third of the world’s food is wasted every year [24], including food loss due to non-subjective factors in production, processing, and storage and food waste due to irrational consumption behavior at the consumption end [25,26]. Li et al. [27] showed that 349  ±  4  Mt food annually produced for human consumption in China was lost or wasted. The impact of such trends on food supply will continue to exist, while relevant supply–demand analysis rarely considers both. It is thus necessary to reveal the balance of food supply–demand in China considering both food for feed, and food loss and waste.

Taking into account the aforementioned issues, our analysis included six major categories of food (including grain crops, oil crops, sugar crops, vegetables and fruits, livestock, and aquatic products) and conducted food production to calorie conversion based on provincial-level data. This study aims to answer the following questions: (1) the spatiotemporal evolution of food production in China at different scales; (2) the dynamics of food supply–demand in China considering food for feed, and loss and waste; (3) the migration of the center of gravity of food “production” and “demand”. The objectives of this study are to further reveal the spatio-temporal evolution characteristics of food production and food self-sufficiency in China and to provide a reference for optimizing the allocation of agricultural resources and achieving a balance between food supply and demand.

## 2. Data and Methods

### 2.1. Data Sources

On the basis of sources such as the China Rural Statistical Yearbook (2009–2020), China Agricultural Statistics (1978–2020), and Statistics on 60 Years of New China’s Agriculture (1978–2008), this study constructed a database on the production of 26 foods in a total of six major categories. The calorie content of the edible part and the proportion of edible parts used for food-yield calorie conversion were obtained from the Chinese Food Composition Table (6th edition) and article [8]. The provincial population data were obtained from the China Statistical Yearbook and the Compilation of Population Statistics of the People’s Republic of China. Feeding percentages were obtained from the China National Cereals and Oils Information Centre (https://www.chinagrain.cn/) (accessed on 10 April 2021) and article [21].

### 2.2. Research Methods

#### 2.2.1. Local Food Supply Estimation

The main uses of grains in China include rations, feed, industry, and seeds, which accounted for about 56.6%, 25.7%, 16.1%, and 1.6%, respectively, in 2015, as published by China’s National Cereals and Oils Information Center. Scholars have used the method of quantifying food for rations by distinguishing among food uses [19]. Considering the two largest proportions, rations and feed grains, as well as the inevitable food loss and waste component, we estimated the local food supply by deducting feed and food loss and waste from the total food production (Equation (1)).
(1)$$Y_{s,t,i}{}^{*}=Y_{s,t,i}\left(1-\gamma_{i}-\lambda_{t}\right)\left(1-\mu \right)$$
where i = 1, 2, 3,..., 26 indicates the food types; s = 1, 2, 3,..., 31 refers to the provinces in mainland China; t = 1978, 1979,..., 2020 indicates the year; $Y_{s,t,i}{}^{*}$ is the local supply of food i in province s in year t; $Y_{s,t,i}$ is the production of food i in province s in year t; $\gamma_{i}$ is the feed-to-production ratio of food i (%), mainly for grain crops (66% for maize, 11% for soybean, 6% for wheat, 5% for rice [21], and zero for others for simplicity); $\lambda_{t}$ is the food loss ratio in year t (%); and μ is the proportion of food wasted (10% was chosen on the basis of article [28]). $\lambda_{t}$ values were found to show a generally decreasing trend [27] of approximately 16.13% in 1978 decreasing to 8.95% in 2020, and were linearly fitted here according to the relevant literature [24,29,30,31,32,33] to characterize the dynamics ($\lambda_{t}$ = −0.1709 t + 354.17, R^2^ = 0.573). Note that the food loss here referred to the post-production loss, including post-harvest handling, storage, processing, and transportation [28].

#### 2.2.2. Food Calorie Yield Model

According to the food calorie yield model [8], we converted food production ($Y_{s,t,i}$) into food calories by multiplying $Y_{s,t,i}$ with the edible portion of each type of food and the calorie content of the edible part at the provincial level, as shown in Equation (2).
(2)$$E_{s,t}=\overset{26}{\underset{i=1}{\sum }}E_{s,t,i}=\overset{26}{\underset{i=1}{\sum }}\left(100\times Y_{s,t,i}\times EP_{i}\times \alpha_{i}\right)$$
where $E_{s,t}$ is the calorie value of food produced in province s in year t (kcal); $E_{s,t,i}$ is the calorie value of food i in province s in year t (kcal); $EP_{i}$ is the proportion of edible portion of food i (%); and $\alpha_{i}$ is the calories per 100 g of edible portion of the food i (kcal). $EP_{i}$ and $\alpha_{i}$ are listed in Table 1. Similarly, we also converted local food supply ($Y_{s,t,i}{}^{*}$) into calories and denoted as $E_{s,t}{}^{*}$, the calorie value of the local food supply in province s in year t (kcal).

#### 2.2.3. Food Calorie Supply–Demand Model

According to the Dietary Nutrient Reference Intakes for Chinese Residents (2017) published by the Chinese Nutrition Society, the daily calorie requirement for an adult of medium weight is 2100–2600 kcal. Here, the average calorie requirement of 2350 kcal/person/day is used to calculate the population that can potentially be fed based on the local food calories (Equation (3)), and the food self-sufficiency is analyzed by comparing the feedable population with the actual population of the year [14,18] (Equation (4)).
(3)$$P_{s,t}{}^{*}=\frac{E_{s,t}{}^{*}}{2350\times 365}$$
(4)$$\Delta P_{s,t}=P_{s,t}{}^{*}-P_{s,t}$$
where $P_{s,t}{}^{*}$ is the feedable population based on food calories provided by the local food supply in province s in year t (persons); $P_{s,t}$ is the actual population in province s in year t (persons); $\Delta P_{s,t}$ is the population gap between the feedable population and the actual population in province s in year t (persons), with a positive value meaning that there is a surplus of local food supply and a negative value meaning that food demand cannot be fully met.

#### 2.2.4. Spatio-Temporal Evolution Analysis

We analyzed the spatio-temporal evolution of food calorie production and the supply–demand equilibrium at four scales: national, production and marketing sub-regions, physical geographical divisions, and provinces.

The division of production and marketing sub-regions is mainly based on the official document “Opinions of the State Council on Further Deepening the Reform of Grain Circulation System” (2001), which divides the 31 provinces in mainland China into the Main Producing Region, the Main Marketing Region, and the Balanced Production–Marketing Region according to the characteristics of grain production and consumption in each province [34] (Figure 1a). Among them, the Main Producing Region (hereinafter referred to as MP) includes Heilongjiang, Jilin, Liaoning, Inner Mongolia, Hebei, Henan, Shandong, Jiangsu, Anhui, Jiangxi, Hubei, Hunan, and Sichuan; the Main Marketing Region (MM) includes Beijing, Tianjin, Shanghai, Zhejiang, Fujian, Guangdong, and Hainan; while the remaining 11 provinces are classified as the Balanced Production–Marketing Region (BPM). Seven physical geographic divisions include (Figure 1b) ① North China (NC): Beijing, Tianjin, Hebei, Shanxi, Inner Mongolia; ② Northeast China (NEC): Heilongjiang, Jilin, Liaoning; ③ East China (EC): Shanghai, Jiangsu, Zhejiang, Anhui, Jiangxi, Shandong, Fujian; ④ Central China (CC): Henan, Hubei, Hunan; ⑤ South China (SC): Guangdong, Guangxi, Hainan; ⑥ Southwest China (SWC): Chongqing, Sichuan, Guizhou, Yunnan, Tibet; ⑦ Northwest China (NWC): Shaanxi, Gansu, Qinghai, Ningxia, Xinjiang.

Linear regressions and significance tests were performed on the total food calorie production at the four different scales from 1978 to 2020, and the trends in food calorie production in the 31 provinces were divided into four groups based on the regression coefficient, the *k*-value (i.e., the annual rate of change in food calorie production, 10^12^ kcal/year), and the *p*-value of the significance test: (1) large increase: *k* ≥ 1 & *p* ≤ 0.05; (2) slight increase: 0 < *k* < 1 & *p* ≤ 0.05; (3) continuous decrease: *k* < 0 & *p* ≤ 0.05; (4) no significant trend: *p* > 0.05.

The feedable population and the population gap between supply and demand were calculated at the different scales from 1978 to 2020 to compare food calorie supply and demand between different regions and identify calorie surplus or deficiency.

In addition, we used the geographical center of gravity model (Equations (5) and (6)) to calculate the spatial center of gravity of the food calorie supply and the population from 1978 to 2020, based on provincial data, in order to visually reveal the spatial changes in food calorie supply and the supply–demand equilibrium.
(5)$$X_{E^{*},t}=\frac{\sum_{s=1}^{31}E_{s,t}{}^{*}\times X_{s}}{\sum_{s=1}^{31}E_{s,t}{}^{*}},Y_{E^{*},t}=\frac{\sum_{s=1}^{31}E_{s,t}{}^{*}\times Y_{s}}{\sum_{s=1}^{31}E_{s,t}{}^{*}}$$
(6)$$X_{P,t}=\frac{\sum_{s=1}^{31}P_{s,t}\times X_{s}}{\sum_{s=1}^{31}P_{s,t}},Y_{P,t}=\frac{\sum_{s=1}^{31}P_{s,t}\times Y_{s}}{\sum_{s=1}^{31}P_{s,t}}$$
where $\left(X_{E^{*},t},Y_{E^{*},t}\right)$ are the coordinates of the geographical center of gravity of the national food calorie supply in year t; $\left(X_{P,t},Y_{P,t}\right)$ is the geographic centre of gravity of the national population in year t; $\left(X_{S},Y_{S}\right)$ is the geographic centre of gravity of the province, which is constant over time; t = 1978, 1979,..., 2020.

## 3. Results

### 3.1. Trends in Regional Food Calorie Production

Food calories produced nationwide in the past 40 years showed a significant upward trend (Figure 2a), increasing from 819.85 × 10^12^ kcal in 1978 to 2172.07 × 10^12^ kcal in 2020, with a growth rate of about 31.7 × 10^12^ kcal/year. The trend in the food calorie production varied among different production and marketing regions (Figure 2b). The food calories produced in both MP and BPM showed a significant increasing trend (*p* < 0.001), with a growth rate of 24.73 × 10^12^ kcal/year and 7.59 × 10^12^ kcal/year, respectively, in contrast to a decreasing trend (−0.60 × 10^12^ kcal/year) in MM. Specifically, the production in MP increased from 544.39 × 10^12^ kcal (66.40% of the total) in 1978 to 1597.39 × 10^12^ kcal (73.54%) in 2020, while the production in MM, which consists of seven economically fast developing provinces, was 153.29 × 10^12^ kcal in 1978 (18.70%), and 162.23 × 10^12^ kcal in 2020 (7.47%). The total food calorie production in MP is making a greater contribution to securing the national food supply.

The total amount of food calories produced by each of the seven physical geographical divisions showed a significant increasing trend (*p* < 0.001, Figure 2c). EC had the strongest food calorie production capacity, indicated by the largest total amount of 618.87 × 10^12^ kcal in 2020, accounting for more than a quarter of the national total, while NWC presented the lowest food calorie production capacity, at 126.54 × 10^12^ kcal in 2020. Considering the growth rate, CC and EC exhibited the fastest development, reaching 7.63 × 10^12^ kcal/year and 7.24 × 10^12^ kcal/year, respectively, while the growth rate for NWC was only 2.03 × 10^12^ kcal/year.

### 3.2. Calorie production in Different Food Categories

At the national level, grain crops had remained the major source of food calories, followed by livestock, and vegetables and fruits (Figure 3a). However, the calorie contribution of grain crops decreased from 90.41% in 1978 to 69.34% in 2000, while that of vegetables and fruits and that of livestock significantly increased. The calorie contributions from different food categories had stabilized since the beginning of the 21st century, with grain crops accounting for 65.52% of the total calories by 2020, and livestock, vegetables and fruits, oil crops, sugar crops, and aquatic products accounting for 11.41%, 10.82%, 7.19%, 3.66%, and 1.40%, respectively.

Regional differences in the production of the six food categories were also found among the different production and marketing sub-regions and physical geographical divisions (Figure 3b). Taking the year 2020 as an example, MP provided 80.52% of the total calories from the grain crops, 78.11% of oil calories, 6.10% of sugar calories, 57.0% of calories from vegetables and fruits, 70.44% from the livestock, and 52.92% from the aquatic products. For food categories other than the sugar crops, MP provided the largest percentages of calories, BMP provided 80.68% of calories from the sugar crops, while MM accounted for 38.45% of aquatic calories and no more than 20% of calories from the rest. Among the physical geographical divisions, EC and CC provided more than half of the country’s grain calories (788.63 × 10^12^ kcal), CC and SWC provided about 67.42% of the total oil calories (87.12 × 10^12^ kcal), and 75.94% of the country’s sugar calories were produced in SC.

### 3.3. Provincial Food Calorie Production

At the provincial level, Jiangsu, Guangdong, Hubei, Hunan, and Sichuan were the major food calorie producing provinces in 1978 (Figure 4a). By 2020, the major producing provinces had shifted northwards to provinces such as Heilongjiang, Shandong, and Henan (Figure 4b), showing a distribution pattern of more in the east and less in the west. Regarding the type of food caloric growth rate (Figure 4c), most provinces presented a significant increase in food calorie production (a large or a slight increase), and the provinces with large increases were mostly located in the eastern region of China, while three provinces, Beijing, Shanghai, and Zhejiang, were in a continuous decrease.

### 3.4. Local Food Calorie Supply and Demand

With the steady increase in the total food calorie production, the flow of food calorie production in China has evolved as shown in Figure 5a. The share of food loss and waste, although decreasing, still accounted for nearly one fifth of the total food calorie production. Without taking into account the import and export trade, China’s food production began to show a calorie surplus in 1992 and could potentially meet the calorie needs of 1.92 billion people by 2020 (Figure 5b). At the levels of the production and marketing sub-regions (Figure 6a) and the physical geographic divisions (Figure 6b), the food calorie supply–demand situation had improved in most regions except for MM and NC, where the supply–demand situation was still not optimistic in 2020. The MP had been in a phase of large surpluses in food calorie production since 1989, with a food surplus equivalent that would have satisfied over 600 million people in 2020, while 150 million people in MM were still dependent on food supplies from other regions. The lower the level of food calorie self-sufficiency, the higher the technical and equipment requirements for food distribution and storage as well as the level of losses and waste. EC and SC were the first to change from a calorie deficit to a surplus, and NWC, with the lowest calorie production capacity, was the last to realize a calorie surplus (in 2011), yet NC was consistently in short supply.

The increase in the feedable population resulting from improved food production together with the growing population has led to a changing pattern in the food calorie supply and demand gap across the country (Figure 7). In 1978, the whole country was in a state of widespread undersupply, particularly in Sichuan province and the Huang-Huai-Hai Basin. The food supply and demand situation has largely improved as the food production capacity increased across the region. In 2020, there was a clear imbalance in food supply and demand, with five provinces presenting a calorie surplus of over 50 million people and 15 provinces unable to meet their own food calorie needs, including two provinces, Guangdong and Fujian, with serious deficits (affecting more than 25 million people). The trend towards heavy reliance on trade logistics and food storage in the context of carbon-neutrality targets and pandemic prevention and control is not conducive to the coordinated and sustainable development of regional industries.

During the period 1978 to 2020, the center of gravity for the actual population moved 63.75 km to the southwest (Figure 8a), while the center of gravity for calorie supply moved 204.67 km to the northeast, shifting from Hubei to Henan province in 2020 (Figure 8b). Such spatially reversed movements of “supply” and “demand” (Figure 8c) will exacerbate the pressure on soil and water resources in the north and further increase the need for efficient logistics and trade networks to guarantee sufficient food supply.

## 4. Discussion

Taking provinces as the basic unit, we unified six food categories including grain crops, oil crops, sugar crops, vegetables and fruits, livestock, and aquatic products with calories as a quantitative scale, and we presented the spatio-temporal food calorie production from 1978 to 2020 at four scales including national, production and marketing sub-regions, physical geographic divisions, and provinces. We further revealed the spatial and temporal evolution of the local food calorie supply–demand equilibrium and illustrated the spatial migration of the center of gravity, taking into account the increasing use of food for feed, and food loss and waste.

From the perspective of food production, China’s total calorie output increased linearly from 1978 to 2020, at a rate of 31.7 × 10^12^ kcal/year, with grain crops contributing the most. The fluctuation of food production in China from 1998 to 2003 might be attributed to the blind implementation of the “Returning Farmland to Forest and Grass” project in many areas [35] and also related to natural disasters such as the 1998 Yangtze River flood. All production and marketing sub-regions (excluding MM), all physical geographical divisions, and the majority of provinces (except Beijing, Shanghai, Zhejiang, and two other provinces) also displayed significant growth rates for food production. Both food calories and caloric growth rates show a distribution pattern of high in the east and low in the west. Improvements in national macro-control, scientific and technical inputs, and other aspects are all necessary to increase the food production capacity. The promotion of technologies such as water-saving irrigation and scientific fertilization has increased the pesticide and fertilizer utilization efficiency and decreased the incidence of pests, illnesses, and weeds. Since the implementation of reform and opening up as well as the household contract responsibility system in 1978, farmers have gained autonomy in food production and distribution, which has greatly motivated farmers and enhanced grain production. China has gradually started to scale up and standardize livestock and poultry farming since 2000 [36]. The government has introduced various policies to support the development of livestock farming, including large-scale construction and compensation for grassland incentives in pastoral areas. In this context, the feed industry also gained rapid development [22]. After a large reduction in food production from 1998 to 2003, China’s “No. 1 Document” has focused on agriculture every year since 2004, giving a strong impetus to food production. With the rapid economic growth, the financial and material support for agriculture has been gradually increased. Apart from guiding the establishment of regional agricultural layout optimization measures such as functional grain production zones and important agricultural production reserves, the state also introduced the Anti-Food Waste Law in 2021. China’s efforts to ensure food security have been ongoing.

With the advancement of urbanization, secondary and tertiary industries in Beijing and Shanghai are rapidly developing, and the trend of “non-agriculturalization” is obvious, resulting in the transfer of agricultural production components such as labor force, land, and water resources [9]. The center of gravity of China’s agricultural labor force has generally shifted to the northwest as a result of the significant influx of rural laborers to the economically developed eastern areas and the hollowing out of the rural labor force [37]. The virtual water flow of domestic food trade from irrigated agriculture-based provinces to non-irrigated agriculture-based provinces and from less economically developed regions to economically developed regions [38], and the migration of the center of gravity of food calories to the north, will have a negative impact on the stability of ecosystems and economic development in the northern regions, where water resources are scarce, and the pattern of “north–south water transfer” should be given high priority. Arable land has also shifted from southern and central China, where the replanting index is higher and the quality of arable land is superior, to northwest and northeast China, where the replanting index is lower and the food production capacity of arable land is generally lower [39], indicating a phenomenon of “taking advantage of the best to compensate for the worst”. In the last 20 years, the arable land in the south has decreased, with urbanization and ecological restoration plans being the main drivers of arable land loss. The trends above will inevitably lead to the evolution of the spatial and temporal pattern of food production, further increasing the pressure on water and soil resources. The timely adjustment of the agricultural cultivation structure and water transfer will be of great significance in alleviating the pressure on regional water and soil resources and ensuring national food security.

From the perspective of food supply and demand, there has been a national food calorie supply surplus since 1992. Zhang [19] did not consider the food loss and waste other than in food crops and concluded that the nation entered the caloric surplus stage in the 1980s, earlier than the results of this study. This study revealed a continuous improvement in the food security situation nationwide for more than 40 years based on the calorie perspective. However, with economic and social development, total food consumption will grow rigidly, and the structure of food consumption will be upgraded, and food consumption for feed and industrial conversion will continue to increase. Food calorie production will continue to be concentrated in core production areas, cross-regional food flows will increase, and the risk of significant market fluctuations will remain. Therefore, the commitment to ensuring national food security cannot yet be relaxed, and the initiative of establishing a big food concept still needs to be promoted to ensure the effective supply of various types of food such as meat, vegetables, fruits, and aquatic products while ensuring the supply of food. The No. 1 Document of the Central Government in 2022 calls for “continuously improving comprehensive production capacity in the Main Production Region, increasing food self-sufficiency in the Main Marketing Region, and ensuring basic food self-sufficiency in the Balanced Production and Marketing Region”. Moreover, this study found that food production trends in MM were still in deficit in 2020 and suggests that food production strategies for MM be developed in terms of resource endowments and cultivation traditions in order to meet the requirement of the government. It is imperative to implement the dual circulation strategy with the domestic circulation as the main element while domestic and international circulations mutually promote each other. Moreover, North China and the 15 provinces with calorie deficiency should also optimize their cultivation structure, improve their food production capacity, establish an efficient and swift distribution system, promote in-depth food saving and loss reduction, and strengthen food security education to reduce food waste.

The center of gravity of food in China is moving to the northeast, while the center of gravity of population is moving to the southwest, and the distance between the earlier and current centers of gravity of calories (204.67 km) is greater than the distance between the earlier and current centers of gravity of population (63.75 km). The trend of the two centers of gravity moving in opposite directions is an important reflection of the contradiction between food supply and demand. There is an urgent need to establish a more efficient and rapid distribution system to fill the gap between supply and demand, with production and distribution complementing each other. Since food loss and waste and greenhouse gas emissions will inevitably occur in the process of storage and transportation, the requirements for storage facilities and logistics will be more stringent. In addition, serious food waste is endangering China’s food security, resource and environmental security, and the nutrition and health of the population. Within a reasonable and achievable range, reducing food loss and waste will significantly reduce the gap between food supply and demand, and there is an urgent need to strictly implement anti-waste policies to ensure the safety and sustainable development of regional food production.

Limited by the availability of research data and historical information, this study still suffers from certain limitations: (1) The wide variety of food items across the country and the differences in the main food sources in different regions constrained comprehensive calculations. However, this study has included 26 food items in six major food categories, which are well representative of the main food items in the country and can largely reveal the spatial and temporal patterns of food calorie production. Note that the study did not take into account food imports, exports, and inter-provincial trade, and focused on demonstrating the food self-sufficiency based on the local food calorie supply and demand balance. (2) Owing to the complexity of the food supply chain system, the diversity of food losses and waste at each stage, and the difficulty in obtaining relevant data, the quantification of food losses and waste remains a challenge. This study conducted a simplified estimation by assuming a steady proportion of food wasted and fitting a historical trend for food loss ratio, which can help reflect the general trend over a period of more than 40 years and is useful for improving the understanding of the spatial and temporal evolution of food calories.

Optimizing food production systems is essential for balanced nutrition, as many countries are experiencing malnutrition due to micronutrient deficiencies. Micronutrient deficiencies still exist in China. For example, the dietary fiber intake of 10.4 g for Chinese residents aged 2 years and older (2015–2017) was lower than the 17.5 g of US residents and the 19.8 g of the Dutch [40]. Folate supply has been considered insufficient to meet current needs [41], and other micronutrients are also important in food security. Future work could further consider the supply and demand of micronutrients to make a more comprehensive evaluation. In addition, future research is needed to predict food calorie production capacity in the context of climate change as well as to quantify the impact of the combined effects of climate, agricultural production, and socio-economic factors on the spatial and temporal evolution of calories. In order to reveal more scientifically the food supply capacity of China in complex international and domestic situations, the above-mentioned issues will be further implemented in future research work.

## 5. Conclusions

From the perspective of food production, China’s total calorie output increased linearly from 1978 to 2020, at a rate of 31.7 × 10^12^ kcal/year, with grain crops contributing the most. Both food calories and caloric growth rates show a distribution pattern of high in the east and low in the west. The current achievements are due to the national macro-control, science and technology investment, and other improvements. However, the spatial and temporal evolution trends of the region are not promising, and the migration of the center of gravity of food production will further increase the pressure on soil and water resources. The timely adjustment of the agricultural cultivation structure and of water transfer are needed to cope with possible future dilemmas.

From the perspective of food supply and demand, there has been a national food calorie supply surplus since 1992, but the supply and demand situation in the Main Marketing areas and Northern China is deteriorating. Food calorie production will continue to be concentrated in core production areas, and the two centers of gravity, “people” and “food”, will move in opposite directions, leading to increased cross-regional food flows; additionally, the risk of large market fluctuations still exists. Therefore, there is an urgent need to establish an efficient and effective distribution system to promote food conservation and loss reduction.

## Figures and Tables

**Figure 1 foods-12-00956-f001:**
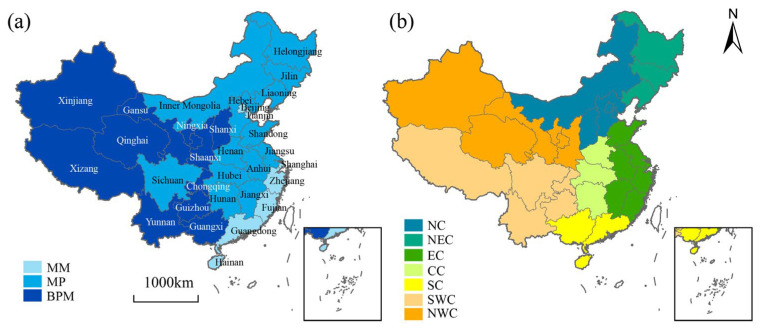
Schematic diagram of division: (**a**) production and marketing sub-regions; (**b**) physical geographic divisions.

**Figure 2 foods-12-00956-f002:**
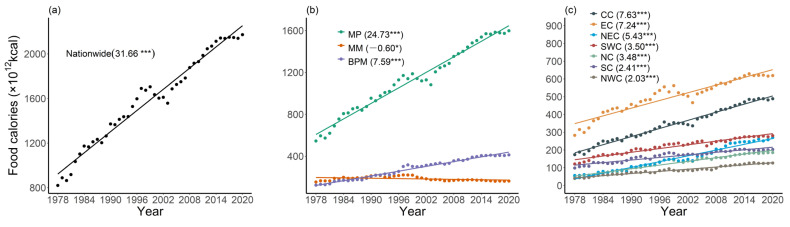
Trends in regional food caloric production from 1978 to 2020: (**a**) nationwide; (**b**) production and marketing sub-regions; (**c**) physical geographic divisions. Note the values in the brackets indicate the growth rate of the food calories (unit: 10^12^ kcal/year), with the signs referring to the significance level (***: *p* < 0.001; *: *p* < 0.05).

**Figure 3 foods-12-00956-f003:**
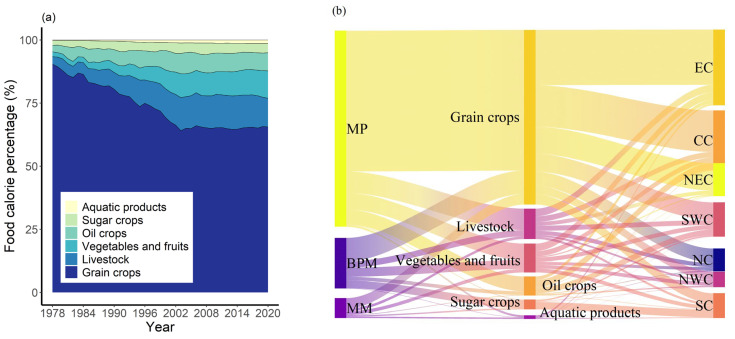
Food caloric production in different categories of food: (**a**) food calorie contribution for different categories of food from 1978 to 2020; (**b**) calorie production of six categories of food in production and marketing sub-regions and physical geographic divisions in 2020.

**Figure 4 foods-12-00956-f004:**
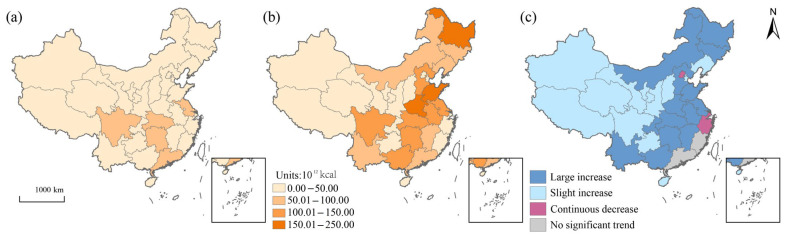
Distribution pattern of food production and classification in China: (**a**) food calorie production pattern in 1978; (**b**) food calorie production pattern in 2020; (**c**) classification of food caloric growth rate from 1978 to 2020.

**Figure 5 foods-12-00956-f005:**
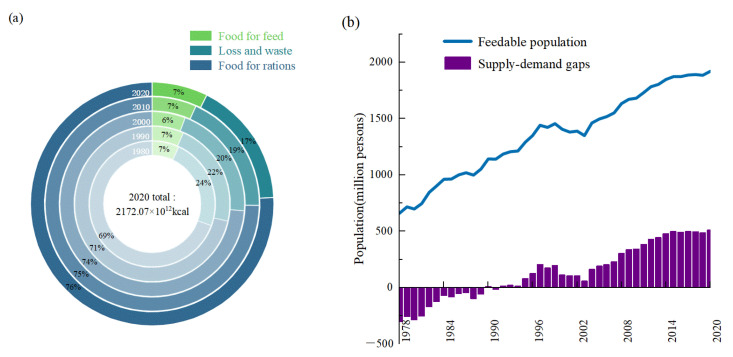
The proportion of each flow direction of food calories and the evolution of supply–demand in China: (**a**) the proportion of each flow direction of food calories; (**b**) the feedable population and supply–demand gaps across the country over time.

**Figure 6 foods-12-00956-f006:**
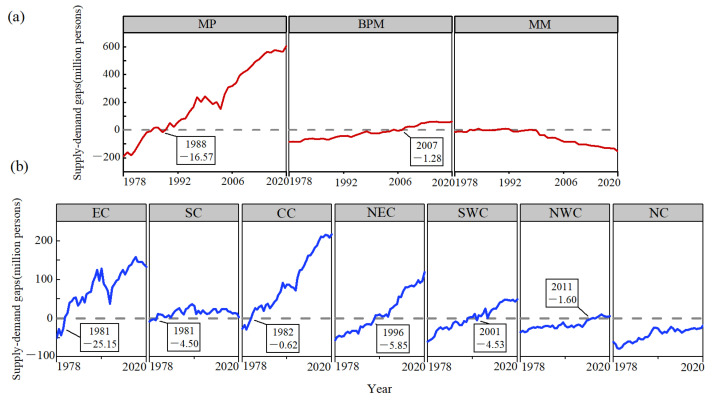
Food caloric supply–demand gap changes in (**a**) production and marketing sub-regions and (**b**) physical geographic divisions. The box marks the last year when the supply deficiency occurred and its deficiency (million persons).

**Figure 7 foods-12-00956-f007:**
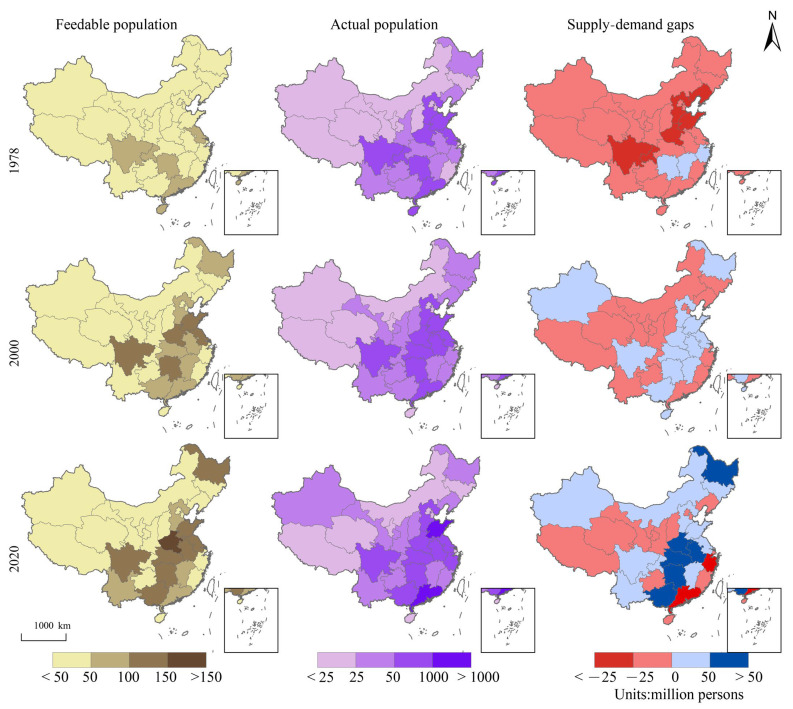
Spatial distribution of food calorie supply–demand equilibrium in China, 1978, 2000, and 2020.

**Figure 8 foods-12-00956-f008:**
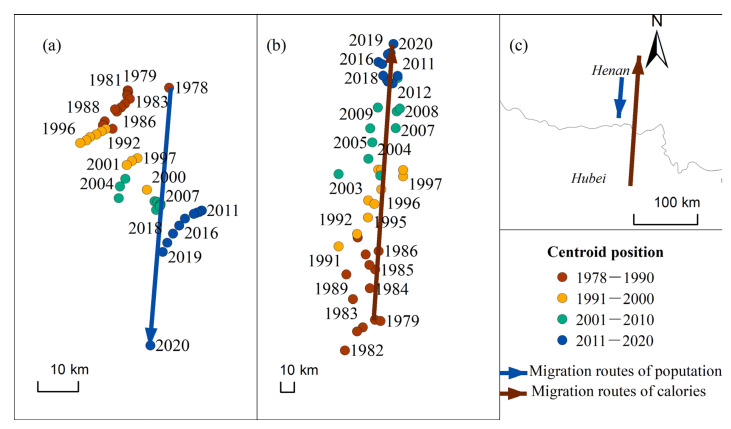
Migration pattern of the centers of gravity of the actual population and food calorie supply from 1978 to 2020: (**a**) population; (**b**) calorie supply; (**c**) trajectory comparison.

**Table 1 foods-12-00956-t001:** Food composition.

Category	Food	*EP* (%)	α (kcal)	Category	Food	*EP* (%)	α (kcal)
grain crops	rice	100	346	oil crops	peanut	53	313
	maize	46	112		rapeseed	100	499
	tuber	92	93	sugar crops	sugar cane	100	64
	wheat	100	338		beet	95	55
	soybean	100	390	vegetables and fruits	vegetables	93	24
livestock	beef	100	160		citrus	77	44
	mutton	100	139		grape	86	45
	eggs	87	167		pineapple	68	44
	honey	100	65		persimmon	87	74
	pork	91	331		apple	85	53
	poultry	65	210		pear	82	51
	milk	100	321		banana	59	93
aquatic products	mean	51	91		red dates	87	125

## Data Availability

All the source data used in this study are publicly available and open access.

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
