# Peer review of "The Spatio-Temporal Evolution of Food Production and Self-Sufficiency in China from 1978 to 2020: From the Perspective of Calories"

_foods, 2023, doi:10.3390/foods12050956_

Round 1
Reviewer 1 Report
The article is a serious study in the field of food security based on the materials of China. Note: It is necessary to explain whether the mathematical dependence (1) was proposed by the authors and to justify in more detail the accounting of food losses.
Due to the fact that the term "food security" is repeated many times in the article, we can recommend its abbreviated name. The final decision on this issue should be made by the editors.
Author Response
Thanks very much for taking your time to review this manuscript (ID: foods-2152010). We really appreciate all your comments and suggestions! These comments are all valuable and helpful for improving our article. According to the comments and suggestions, we have made extensive modifications to our manuscript to make our results convincing. Please find our itemized responses below and our revisions in the re-submitted files.
Point :
The article is a serious study in the field of food security based on the materials of China. Note: It is necessary to explain whether the mathematical dependence (1) was proposed by the authors and to justify in more detail the accounting of food losses.
Due to the fact that the term "food security" is repeated many times in the article, we can recommend its abbreviated name. The final decision on this issue should be made by the editors.
Response:
We are very grateful for your comments on the manuscript. According to your advice, we amended the part in the manuscript.
- 1. Equation 1 is to estimate the local food supply by deducting feed and food loss and waste from the total food production. The method of obtaining the amount of food supplied for direct human intake by differentiating food uses has been used by scholars [19]. Equation 1 was chosen to further quantify the local food supply in a more refined way based on previous studies, for example, by quantifying food for feed through the feed ratio of each grain crop rather than the grain consumption coefficient of each livestock product, and by fitting the loss ratio to obtain the changing food loss.
We have added the following description in Lines 134-135 to clarify this:
"Scholars have used the method of quantifying food for rations by distinguishing among food uses [19]. "
- 2. Your question is very good and reasonable. The quantification of food loss remains a challenge, due to the complexity of the food supply chain system, the diversity of food loss at each stage, and the difficulty in obtaining relevant data,. And the proportion of food loss in China has changed during the development of more than 40 years. To address this issue, we have summarized the relevant data for estimation based on the existing literature.
We carefully screened nationwide specific data [24,29-33] that fit the post-production loss measure (post-production loss including post-harvest handling, storage, processing and transportation [27]) and fitted the trend with reference to the decreasing trend of food loss proportion in the development process of developed countries [28]. The resulting linear equation is shown below, which can reflect the relevant trends to some extent. The fitting results showed that the food loss ratio in 1978 was about 16.13% and decreased to 8.95% in 2020.
Considering that the detailed expansion of this section in the article takes up space and may mislead the reader's understanding of the key points of the article, the estimation of food loss in the article is summarized as " values were found to show a generally decreasing trend[28], about 16.13% in 1978 and decreasing to 8.95% in 2020, and were linearly fitted here according to the relevant literature[24,29-33] to characterize the dynamics ( = -0.1709t + 354.17, R2 = 0.573). Note the food loss here referred to the post-production loss, including post-harvest handling, storage, processing and transportation [27]. " Please see page 3 of the revised manuscript, line 144-150.
Hope that the above explanations will make our article clearer now. We would like to thank the refers again for taking the time to review our manuscript.

Reviewer 2 Report
The paper presents a comprehensive temporal-spatial evaluation of food production in China between 1978 and 2020. In general, the paper is well organized and well written, and the method, data, and results are well and clearly presented. The results obtained are meaningful for the policy-makers to better formulate future policies and schemes for agricultural development in different countries and regions of China. Furthermore, the limitations of the current work are also clearly pointed out. Thus, I believe the paper can be accepted for publication, and the following recommendations may help for a better presentation of the research.
1, Overall, the readability of the paper is very good, but minor language improvements may be done. For example, the use of over-long sentences needs to be minimized.
2, The logical connections in the Introduction need to be better justified. For example, how food productivity and supply will help achieve the carbon neutrality goal? Page 1, line 42-43.
3, Has similar research been conducted? A literature review may help better justify the literature gaps. What are the research questions the paper aims to answer? These may be added in the Introduction.
4, Page 13, line 119, the data used is from 2015, can it be updated with more recent data?
5, Section 2.2.4, even if detailed information is given, the presentation here needs to be improved. For example, a map with the provinces and the regions defined in lines 176-181 will help the readers to better follow the paper.
6, Figure 1, the linear regression shows a good fit with the data. Nevertheless, large fluctuations are observed during 1996-2005, how this may be explained?
7, The last section needs to be the conclusion (not the discussion), which includes a summary of the paper, a discussion of the results, limitations, and future work. Limitations and future works are not the same and need thus to be given separately.
8, Finally, apart from these, are there any insights that may be related to food imports in China? As China is becoming one of the largest food importers in the world.
Author Response
We are very grateful for your comments on the manuscript. According to your advice, we have made further revisions in the manuscript. All of your comments and suggestions were answered one-by-one.
Point 1: The paper presents a comprehensive temporal-spatial evaluation of food production in China between 1978 and 2020. In general, the paper is well organized and well written, and the method, data, and results are well and clearly presented. The results obtained are meaningful for the policy-makers to better formulate future policies and schemes for agricultural development in different countries and regions of China. Furthermore, the limitations of the current work are also clearly pointed out. Thus, I believe the paper can be accepted for publication, and the following recommendations may help for a better presentation of the research.
1, Overall, the readability of the paper is very good, but minor language improvements may be done. For example, the use of over-long sentences needs to be minimized.
Response 1: Thank you for reading carefully and giving positive comments. As suggested, we have checked through the whole manuscript and revised the over-long sentences.
Detailed changes can be found in the revised manuscript in lines 49-54, lines 84-104, lines 109-115, and lines 427-430.
Point 2: The logical connections in the Introduction need to be better justified. For example, how food productivity and supply will help achieve the carbon neutrality goal? Page 1, line 42-43.
Response 2: Thank you for your generous comments and careful check. According to your suggestion, we have modified the relevant text in Lines 42-46 on Page 1:
“Within the context of the current carbon-neutral development goals and the post-epidemic era, improving food productivity and ensuring sufficient food supply can facilitate efficient use of natural resource endowments, rationalize the layout of food production, and reduce unnecessary emissions and resource waste.”
Point 3: Has similar research been conducted? A literature review may help better justify the literature gaps. What are the research questions the paper aims to answer? These may be added in the Introduction.
Response 3: We sincerely appreciate the valuable comments. We have checked the literature carefully and added more relevant descriptions in the Introduction of the revised manuscript.
Firstly, analysis of food production and supply and demand from a caloric perspective has become a widely used method. As suggested, we have further reviewed the relevant articles and justified the research gap.
Specifical modifications are shown below:
"Analysis of production and supply and demand in terms of calories has become a widely used method. Beltran et al [14] analyzed global food supply and demand and projected future scenarios through calorie conversion of major grain crops. Tilman [15] calculated the net global demand for crops in terms of calories and protein. However, these large-scale studies generally deal only with major grain crops, which is inadequate since the diet of the population has become more abundant. " (Page 2 Line 66-71)
“Huang et al.[22] evaluated China's feed grain demand and concluded that the feed grain deficit will further increase.” (Page 2 Line 95-96)
“Li et al.[28]showed that 349 ± 4 Mt food annually produced for human consumption in China was lost or wasted.” (Page 3 Line 104-105)
Secondly, we have made the research question more explicit in the introduction section by adding the relevant description:
“This study aims to answer the following questions: 1) the spatiotemporal evolution of food production in China at different scales; 2) the dynamics of food supply-demand in China considering food for feed and loss and waste; 3) the migration of the center of gravity of food ‘production’ and ‘demand’.” (Page 3 Line 112-115)
Point 4: Page 13, line 119, the data used is from 2015, can it be updated with more recent data?
Response 4: Thank you for pointing this out. We also tried to update with more recent data, however, according to the data published by China's National Cereals and Oils Information Center, the year with specific figures is 2015, and data from other sources may lack authority in comparison. Therefore, under comprehensive consideration, we use the data from the year 2015 as evidence that the proportion of food rations and feed rations is relatively large.
Point 5: Section 2.2.4, even if detailed information is given, the presentation here needs to be improved. For example, a map with the provinces and the regions defined in lines 176-181 will help the readers to better follow the paper.
Response 5: Your suggestions take into account the feelings of readers and are the direction of our improvement. In order to facilitate a better reading experience for our readers, we have added relevant maps to show the specific distribution of different regions, as shown below. Please see Page 5 of the revised manuscript, lines 197-199.
Figure 1 Schematic diagram of division. (a) Production and marketing sub-regions; (b) Physical geographic divisions.
Point 6: Figure 1, the linear regression shows a good fit with the data. Nevertheless, large fluctuations are observed during 1996-2005, how this may be explained?
Response 6: Thank you for your careful attention. To be more precise China's nationwide food calorie production was in a fluctuating or even negative growth phase from 1998 to 2003. During this period of time, the main factor that led to the serious decline of food was probably due to the blind implementation of the project of "Returning Farmland to Forest and Grass" in many areas of China. It may also be closely related to natural disasters such as the 1998 Yangtze River flood.
In the Discussion part, we give a brief explanation of this issue "The fluctuation of food production in China from 1998 to 2003 might be attributed to the blind implementation of the "Returning Farmland to Forest and Grass" project in many areas[35] and also related to natural disasters such as the 1998 Yangtze River flood.” (Page 11 Line 366-368)
[35]Yang L. Analysis of Grain Yield Forecast Based on Intervention Model. Advances in Social Sciences, 2017, 06(07),955-963.
Point 7: The last section needs to be the conclusion (not the discussion), which includes a summary of the paper, a discussion of the results, limitations, and future work. Limitations and future works are not the same and need thus to be given separately.
Response 7: We agree with the suggestions about distinguishing between conclusions and discussion, limitations, and future work. We, therefore, added a conclusion part and described the limitations separately from the future work.
Limitations:” Limited by the availability of research data and historical information, this study still suffers from certain limitations: (1) The wide variety of food items across the country and the differences in the main food sources in different regions constrained comprehensive calculations. However, this study has included 26 food items in six major food categories, which are well representative of the main food items in the country and can largely reveal the spatial and temporal patterns of food calorie production. Note the study did not take into account of food imports, exports, and interprovincial trade, and focused on demonstrating food self-sufficiency based on the local food calorie supply and demand balance. (2) Due to the complexity of the food supply chain system, the diversity of food losses and waste at each stage, and the difficulty in obtaining relevant data, the quantification of food losses and waste remains a challenge. This study conducted a simplified estimation by assuming a steady proportion of food wasted and fitting a historical trend for food loss ratio, which can help reflect the general trend over a period of more than 40 years and is useful for improving the understanding of the spatial and temporal evolution of food calories.” (Page 13 Line 466-480)
Future work:” Optimizing food production systems is essential for balanced nutrition, as many countries are experiencing malnutrition due to micronutrient deficiencies. Micronutrient deficiencies still exist in China. For example, the dietary fiber intake of 10.4 g for Chinese residents aged 2 years and older (2015-2017) was lower than 17.5 g for US residents and 19.8 g for the Dutch [40]. Folate supply has been considered insufficient to meet current needs [41], and other micronutrients are also important in food security. Future work could further consider the supply and demand of micronutrients to make more comprehensive evaluation. In addition, future research is needed to predict food calorie production capacity in the context of climate change as well as quantify the impact of the combined effects of climate, agricultural production and socio-economic factors on the spatial and temporal evolution of calories. In order to reveal more scientifically the food supply capacity of China under complex international and domestic situations, the above-mentioned issues will be further implemented in future research work. " (Page 13 Line 481-493)
Conclusion:” From the perspective of food production, China's total calorie output increased linearly from 1978 to 2020, at a rate of 31.7×1012 kcal/year, with grain crops contributing the most. Both food calories and caloric growth rates show a distribution pattern of high in the east and low in the west. The current achievements are due to the national macro-control, science and technology investment and other improvements. However, the spatial and temporal evolution trends of the region are not promising, and the migration of the center of gravity of food production will further increase the pressure on soil and water resources. Timely adjustment of agricultural cultivation structure and water transfer are needed to cope with possible future dilemmas.
From the perspective of food supply and demand, there was a national food calorie supply surplus since 1992., but the supply and demand situation in the Main Marketing areas and Northern China is deteriorating. Food calorie production will continue to be concentrated in core production areas, and the two centers of gravity, "people" and "food", will move in opposite directions, leading to increased cross-regional food flows, and the risk of large market fluctuations still exists. Therefore, there is an urgent need to establish an efficient and effective distribution system to promote food conservation and loss reduction.” (Page 13 Line 494-511)
Point 8: Finally, apart from these, are there any insights that may be related to food imports in China? As China is becoming one of the largest food importers in the world.
Response 8: Thank you for your suggestions. This article does not cover food import and export, but focuses more on domestic food production and supply and demand in China, so we don't provide more insights on food import and export. We explained the focus of our study in the introduction and discussion.
(Page 2 Lines 84-90): “Despite the significance of trade for food supply, localization of supply chains has been largely emphasized due to the obstruction and even collapse of the global and regional trade amid the COVID-19 pandemic. Localization supply can also have a positive effect on reducing greenhouse gas emissions during transport and storage [20] considering the carbon neutrality goal. Analyzing local food supply and demand and demonstrating the food self-sufficiency of the region can thus contribute to a better understanding of the current status and trends of agricultural production and regional food supply.”
(Page 13 Lines 471-474): “Note the study did not take into account of food imports, exports, and interprovincial trade, and focused on demonstrating food self-sufficiency based on the local food calorie supply and demand balance.”
We hope that the above explanations will make the content of our article clearer now. We would like to thank the reviewer again for taking the time to review our manuscript.

Reviewer 3 Report
Dear Editor.
I rated the manuscript entitled: "Spatio-temporal evolution of food production and self sufficiency in China from 1978 to 2020: from the perspective of calories". In this article, the temporal and spatial statistical evaluation of calorie intake in China has been investigated.
1- This article has only focused on caloric intake, while other micronutrients are also important in food security. Therefore, it is necessary to consider them as well.
2- Research literature needs substantial strengthening.
3- This is a statistical review paper and its methodology is very simple. In such a situation, it is expected that the effects of important economic and non-economic policies on the improvement of food security in China will be investigated. After doing this, the article should be re-evaluated.
Regards
Author Response
We are very grateful for your comments on the manuscript. All of your comments and suggestions were answered one-by-one.
Point 1: This article has only focused on caloric intake, while other micronutrients are also important in food security. Therefore, it is necessary to consider them as well.
Response 1: This is a good suggestion and it provides a broad perspective on the current state of food production. We agree with the importance of micronutrients for food security, however, the analysis was conducted from the perspective of food calories as the title of the article emphasized. And, the current scope of work in this paper still supports the conclusions. Yet, according to the suggestion, we added the importance of micronutrients and suggest that research on trace elements can be invested in future work, in the Discussion section.
The specific description in the revised version is " Optimizing food production systems is essential for balanced nutrition, as many countries are experiencing malnutrition due to micronutrient deficiencies. Micronutrient deficiencies still exist in China. For example, the dietary fiber intake of 10.4 g for Chinese residents aged 2 years and older (2015-2017) was lower than 17.5 g for US residents and 19.8 g for the Dutch [40]. Folate supply has been considered insufficient to meet current needs [41], and other micronutrients are also important in food security. Future work could further consider the supply and demand of micronutrients to make more comprehensive evaluation." (Page 13 Line 481-488)
[40] Yu D M, Zhao L Y, Ju L H. Status of energy and primary nutrients intake among Chinese population in 2015-2017. Food and Nutrition in China, 2021,27(4),5-10.
[41] Wood S A, Smith M R, Fanzo J. Trade and the equitability of global food nutrient distribution. Nature sustainability, 2018, 1(1), 34-37.
Point 2: Research literature needs substantial strengthening.
Response 2: We sincerely appreciate the valuable comments. We have checked the literature carefully and added more relevant descriptions into Introduction part in the revised manuscript.
Specifically expressed as "Analysis of production and supply and demand in terms of calories has become a widely used method. Beltran et al [14] analyzed global food supply and demand and projected future scenarios through calorie conversion of major grain crops. Tilman [15] calculated the net global demand for crops in terms of calories and protein. However, these large-scale studies generally deal only with major grain crops, which is inadequate since the diet of the population has become more abundant. " (Page 2 Line 66-71)
“Huang et al.[22] evaluated China's feed grain demand and concluded that the feed grain deficit will further increase.” (Page 2 Line 95-96)
“Li et al.[28]showed that 349 ± 4 Mt food annually produced for human consumption in China was lost or wasted.” (Page 3 Line 104-105)
Point 3: This is a statistical review paper and its methodology is very simple. In such a situation, it is expected that the effects of important economic and non-economic policies on the improvement of food security in China will be investigated. After doing this, the article should be re-evaluated.
Response 3: Thank you for your review. As suggested, we have added some explanations on the impact of the policies on China's food security.
“Since the implementation of Reform and Opening-up as well as the Household contract responsibility system in 1978, farmers have gained autonomy in food production and distribution, which has greatly motivated farmers and enhanced grain production. China has gradually started to scale-up and standardize livestock and poultry farming since 2000 [36]. The government has introduced various policies to support the development of livestock farming, including large-scale construction and compensation for grassland incentives in pastoral areas. In this context, the feed industry also gained rapid development [22]. After a large reduction in food production from 1998 to 2003, China's "No. 1 Document" has focused on agriculture every year since 2004, giving a strong impetus to food production. With the rapid economic growth, the financial and material support for agriculture has been gradually increased. Apart from guiding the establishment of regional agricultural layout optimization measures such as functional grain production zones and important agricultural production reserves, the state also introduced the Anti-Food Waste Law in 2021. China's efforts to ensure food security have been ongoing.” (Page11 Line 377-391)
[22] Huang S, Liu A, Lu C. Supply and demand levels for livestock and poultry products in the Chinese mainland and the potential demand for feed grains. Journal of Resources and Ecology, 2020, 11(05),475-482.
[36]Cheng S K, Wang S Y, Liu X, et al. Food nutrition and food security of China in a new development phase. Chin Sci Bull, 2018, 63,1764–1774,
We would like to thank the reviewer again for taking the time to review our manuscript.

Round 2
Reviewer 2 Report
The authors have put significant effort to improve the paper. All my previous comments are well considered in the revision or properly responded to, so I think the paper can be accepted in its present form.
Reviewer 3 Report
Dear Editor. I'm happy about the revised paper and recommend you to accept it.
Regards